# Synthetic cytokine receptors transmit biological signals using artificial ligands

Erika Engelowski[1], Artur Schneider[1], Manuel Franke[1], Haifeng Xu[2], Ramona Clemen[1], Alexander Lang[1], Paul Baran[1], Christian Binsch[3], Birgit Knebel[3], Hadi Al-Hasani[3,4], Jens M. Moll[1], Doreen M. Floß[1], Philipp A. Lang[2] & Jürgen Scheller[1]

Cytokine-induced signal transduction is executed by natural biological switches, which among many others control immune-related processes. Here, we show that synthetic cytokine receptors (SyCyRs) can induce cytokine signaling using non-physiological ligands. High-affinity GFP- and mCherry-nanobodies were fused to transmembrane and intracellular domains of the IL-6/IL-11 and IL-23 cytokine receptors gp130 and IL-12Rβ1/IL-23R, respectively. Homo- and heterodimeric GFP:mCherry fusion proteins as synthetic cytokine-like ligands were able to induce canonical signaling in vitro and in vivo. Using SyCyR ligands, we show that IL-23 receptor homodimerization results in its activation and IL-23-like signal transduction. Moreover, trimeric receptor assembly induces trans-phosphorylation among cytokine receptors with associated Janus kinases. The SyCyR technology allows biochemical analyses of transmembrane receptor signaling in vitro and in vivo, cell-specific activation through SyCyR ligands using transgenic animals and possible therapeutic regimes involving non-physiological targets during immunotherapy.

[1] Institute of Biochemistry and Molecular Biology II, Medical Faculty, Heinrich-Heine-University, 40225 Düsseldorf, Germany. [2] Institute of Molecular Medicine II, Medical Faculty, Heinrich-Heine-University, 40225 Düsseldorf, Germany. [3] Institute for Clinical Biochemistry and Pathobiochemistry, German Diabetes Center, Medical Faculty, Heinrich-Heine-University, 40225 Düsseldorf, Germany. [4] German Center for Diabetes Research, 85764 München-Neuherberg, Germany. These authors contributed equally: Erika Engelowski, Artur Schneider, Manuel Franke. Correspondence and requests for materials should be addressed to J.S. (email: jscheller@uni-duesseldorf.de)

Synthetic biology deconstructs and reassembles biological bits and pieces to construct biological devices for applications such as biological sensors, releasers, and switches[1]. Cytokine-induced signal transduction is executed by natural biological switches among many other functions control immune-related processes[2]. In principle, cytokine receptors are in an off-state in the absence of cytokines and in an on-state in the presence of cytokines. The on-state might be interrupted by negative feedback mechanisms or depletion of the cytokine and cytokine receptor. In the past, we reported ligand-independent synthetic receptors based on fusions of leucine zippers or IL-15/sushi and the IL-6-signal transducer gp130, which are locked in the on-state, but were not switchable[3, 4]. Interestingly, a marked activation of IL-6/IL-11 signaling in inflammatory hepatocellular adenomas was directly caused by gain-of-function mutations within the gp130-receptor chain, leading to ligand-independent constitutively active gp130 receptors[5]. Others described switchable synthetic cytokine receptors, resulting in gp130-induced signaling by stimulation with the cytokine erythropoietin (EPO)[6]. The major drawback of this system was that EPO has cross-reactivity with its natural EPO-receptors limiting its applications both in vitro and in vivo. Also, higher ordered multi-receptor complexes cannot be assembled using natural ligands such as EPO, which only induces receptor-homodimerization. Direct intracellular activation of signal transduction and induction of cell death was achieved using cell permeable, synthetic ligands (FK506), and binding proteins (FKBP12) resulting in homodimerization and homooligomerization[7, 8]. The extent of oligomerization was, however, not controllable. Various formats of synthetic transmembrane receptors have been designed to optimize engineered chimeric antigen receptor (CAR) T-cell responses, including co-stimulatory receptors[9–11], notch-based receptors[12], and antigen-specific inhibitory receptors[13]. However, a switchable and background-free synthetic cytokine receptor system with full control over the assembly modus of the receptor complexes, e.g., hetero/homodimeric, -trimeric, or –multimeric is not available. Such a specific system would be a valuable tool to study receptor activation, their kinetics, stoichiometry, and biochemical properties. Moreover, background-free activation of cytokine receptors opens a great potential for novel therapeutic regimes involving non-physiological ligands during immunotherapy. Recently developed nanobodies specifically recognizing GFP and mCherry fail to bind endogenous ligands[14, 15] and thus qualify as binding partners of synthetic cytokine receptors. The N-terminal region of Camelidae heavy-chain antibodies contains a dedicated variable domain, referred to as VHH or nanobodies, which binds to its cognate antigen. Nanobodies are single-domain antibodies of about 110 amino-acid residues generated from the variable regions of these heavy-chain antibodies[16]. Here, these nanobodies were used as extracellular sensors for homo- and heteromeric GFP:mCherry fusion proteins as part of Synthetic Cytokine Receptors (SyCyRs), ultimately leading to the formation and activation of homo- and heterodimeric and heterotrimeric receptor complexes. As biological read-out system, we use IL-23- and IL-6/IL-11-signaling. Consequently, the extracellular sensors were fused to intracellular IL-23- and gp130-receptor chains. Using this set-up, we design a switchable synthetic cytokine receptor system, which resembles IL-23- and IL-6/IL-11-signaling and reveal that homodimeric IL-23R were biologically active. Moreover, we demonstrate that the Janus kinase activity and STAT3 phosphorylation-binding site in the intracellular domain (ICD) of the receptor can be separated on two different receptor chains, a phenomenon, which is referred to as trans-phosphorylation.

## Results

**SyCyRs simulate IL-23 and IL-6/IL-11 signaling.** Natural biological switches regulate cytokine-induced signal transduction via receptor activation and inactivation. Highly specific nanobodies against GFP and mCherry were selected to mediate sensitive and background-free cytokine-like signaling. Naturally, IL-23 signals via its receptor complex consisting of IL-23R and IL-12Rβ1. To mimic IL-23 signaling, we generated two synthetic cytokine receptors (SyCyRs) in which the extracellular part, including the ligand-binding site of the IL-23R and the IL-12Rβ1 was replaced by nanobodies specifically recognizing mCherry ($C_{VHH}$) and GFP ($G_{VHH}$), respectively[14, 15] (Fig. 1a and Supplementary Fig. 1a). SyCyRs were expressed in Ba/F3-gp130 cells (Supplementary Fig. 1b), which proliferate following STAT3 activation by the fusion protein of IL-6 and soluble IL-6R named Hyper-IL-6 (HIL-6)[17] (Fig. 1b). As synthetic ligands for SyCyRs, we generated GFP-mCherry fusion proteins and various combinations (Fig. 1a and Supplementary Figs. 2 and 3). Since STAT3 activation is a hallmark of IL-23 signal transduction, we tested if STAT3-dependent proliferation of Ba/F3-gp130 cells expressing $C_{VHH}$-IL-23R and $G_{VHH}$-IL-12Rβ1 (Ba/F3-SyCyR(IL-23/2A) generated using 2A-technology with both cDNAs combined in one open reading frame and Ba/F3-SyCyR(IL-23) generated with two separate cDNAs could be induced with GFP and mCherry proteins and fusions of these. Interestingly, proliferation of Ba/F3-SyCyR(IL-23/2A) cells was specifically induced by GFP-mCherry and 2xGFP-mCherry fusion proteins, but not by single GFP and mCherry proteins (Fig. 1b). Ba/F3 cells only expressing $C_{VHH}$-IL-23R or $G_{VHH}$-IL-12Rβ1 failed to proliferate, demonstrating the high selectivity of GFP-mCherry as a synthetic cytokine ligand. Comparative analysis of the dose-dependent proliferation of Ba/F3-SyCyR(IL-23/2A) and Ba/F3-IL-12Rβ1-IL-23R cells with GFP-mCherry and Hyper-IL-23 (HIL-23), respectively, revealed that the natural cytokine and synthetic cytokine exhibited comparable potency with half-maximal proliferation achieved with HIL-23 (56 kDa) and GFP-mCherry (57 kDa) concentrations of about 5–10 ng/ml (Fig. 1c). GFP-mCherry-induced cellular proliferation of Ba/F3-SyCyR(IL-23/2A) cells was specifically inhibited by a soluble $G_{VHH}$–$C_{VHH}$ fusion protein (Fig. 1d and Supplementary Fig. 4). Analysis of intracellular signal transduction showed that the GFP-mCherry, 2xGFP-mCherry fusion proteins, but not single mCherry, GFP and 3xGFP proteins induced JAK, STAT3, ERK1/2, and AKT phosphorylation in Ba/F3-SyCyR(IL-23/2A) cells (Fig. 1e, f). SyCyR(IL-23/2A) also resulted in specific STAT3 activation in U4C cells (Fig. 1g). Kinetics of pSTAT3 and SOCS3 induction following of Ba/F3-SyCyR(IL-23/2A) and Ba/F3-IL-12Rβ1-IL-23R cells with either GFP-mCherry or HIL-23 were comparable (Fig. 2). Since the IL-23R complex is not targeted by SOCS3, activation resulted in sustained STAT3 phosphorylation[18]. Next, we analyzed the mRNA-expression by gene-array analysis of Ba/F3-SyCyR(IL-23/2A) cells stimulated with GFP-mCherry and Ba/F3-IL-12Rβ1-IL-23R cells stimulated with HIL-23. GFP-mCherry and HIL-23 up- or downregulated 107 and 193 genes, respectively, by a factor 1.5 or more (Fig. 3a). Among them are typical STAT3-target genes, including PIM1, SOCS3 and OSM (Fig. 3b, c). Pathway analysis revealed that the genes regulated by GFP-mCherry and HIL-23 belong to the same pathways (Fig. 3d and Supplementary Figs. 5–7). Our data revealed a high degree of overlap between the transcriptome induced by the synthetic ligand as compared to the natural cytokine.

To investigate whether SyCyRs can be activated by homodimeric ligands, we adapted this system to the IL-6/IL-11 receptor complex[19]. A SyCyR for gp130 was generated in which the extracellular part of the cytokine receptor was replaced by $G_{VHH}$ ($G_{VHH}$-gp130) (Fig. 4a and Supplementary Fig. 1a). Expression of

$G_{VHH}$-gp130 in Ba/F3-gp130 cells (Ba/F3-SyCyR(IL-6)) was verified by flow cytometry (Supplementary Fig. 1b). As expected, JAK, TYK, STAT3, ERK1/2 phosphorylation in Ba/F3-SyCyR(IL-6) cells was specifically induced by 2xGFP-mCherry and 3xGFP (Fig. 4b, c). Comparison of the dose-dependent proliferation of Ba/F3-SyCyR(IL-6) and Ba/F3-gp130 with 3xGFP and HIL-6, respectively, showed that the natural and synthetic cytokine exhibited similar activity with half-maximal proliferation at 1–10 ng/ml 3xGFP (86 kDa) or HIL-6 (60 kDa), respectively (Fig. 4d).

As depicted in Fig. 4e, 3xGFP-stimulation of Ba/F3-SyCyR(IL-6) cells resulted in time-dependent fast activation and slight inactivation of STAT3 phosphorylation after 120 min, which was accompanied by upregulation of SOCS3. Overall, 3xGFP-induced signal transduction was undistinguishable from HIL-6 induced STAT3 phosphorylation and SOCS3 expression. Next, we expressed SyCyR(IL-6) in liver tissue of C57BL/6 mice. Twenty-four hours after injection of cDNAs coding for $G_{VHH}$-gp130 and 3xGFP alone or in combination, we observed STAT3

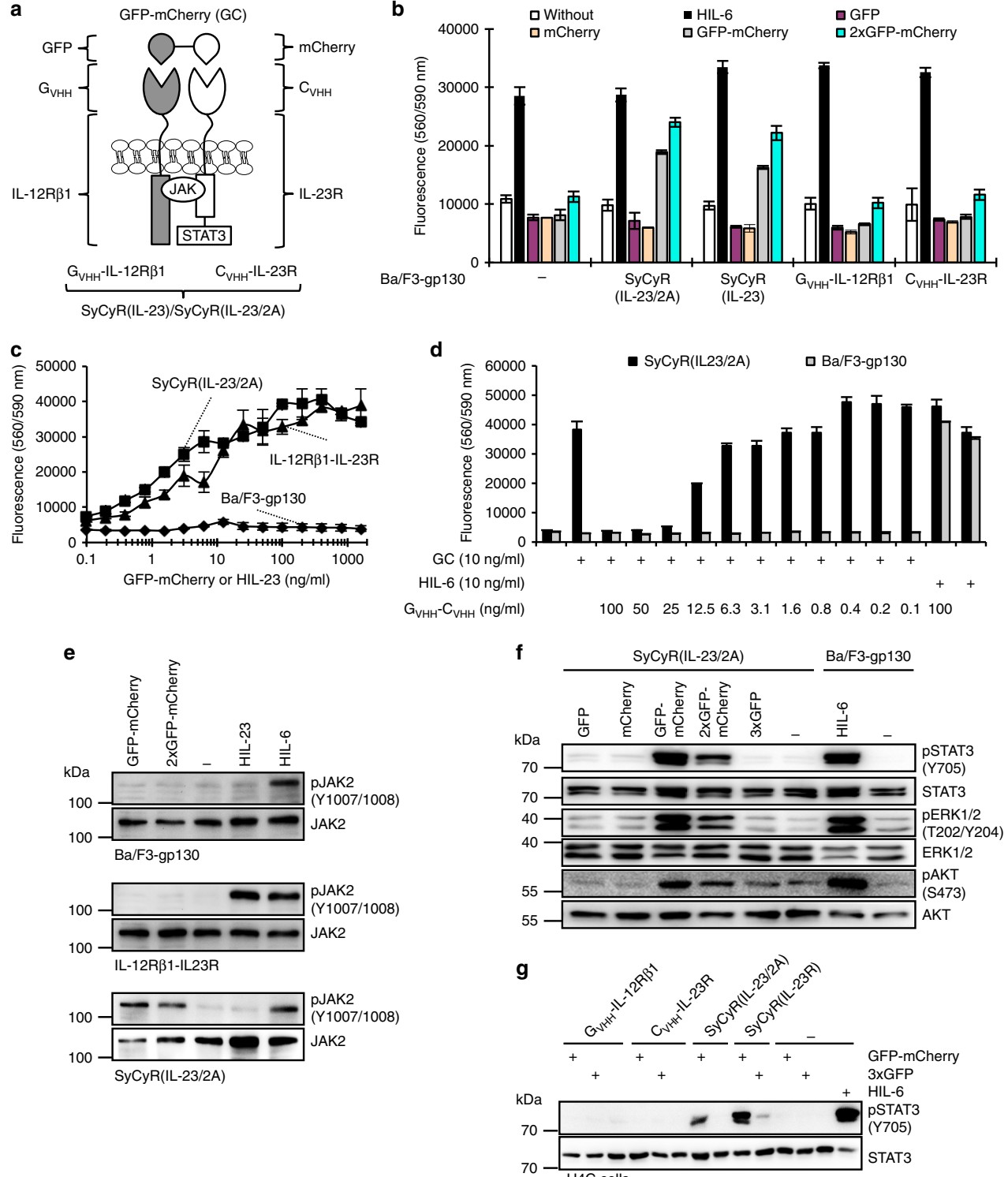

phosphorylation when $G_{VHH}$-gp130 was coexpressed with 3xGFP (Fig. 4f, g). Interestingly, $G_{VHH}$-gp130 expression was found to be much higher in mice injected only with the cDNA coding for $G_{VHH}$-gp130 as compared to mice expressing both, $G_{VHH}$-gp130 and 3xGFP. The data suggest that co-expression of $G_{VHH}$-gp130 and 3xGFP resulted in activation-dependent degradation of $G_{VHH}$-gp130. Moreover, detection of 3xGFP in serum samples by western blotting showed strong accumulation of 3xGFP in mice injected only with the cDNA coding for 3xGFP, whereas 3xGFP was hardly detectable in mice injected with cDNAs coding for $G_{VHH}$-gp130 and 3xGFP. These findings suggest that not only $G_{VHH}$-gp130 but also 3xGFP protein was efficiently internalized and degraded in liver cells after binding and activation of $G_{VHH}$-gp130 (Fig. 4f). Consistently, expression of the acute phase response gene *Saa1* was increased following injection of cDNAs coding for $G_{VHH}$-gp130 and 3xGFP (Fig. 4h), in sharp contrast to injection of cDNA coding for $G_{VHH}$-gp130 or 3xGFP alone. Overall, our data showed that the tested SyCyRs phenocopied IL-6 and IL-23 signaling in vitro and in vivo.

**IL-23R-SyCyR homodimers are biologically active.** Since multimeric GFP was able to induce IL-6 signal transduction via homodimeric $G_{VHH}$-gp130, we wondered, whether IL-23R is also biologically active as a homodimer. Accordingly, we created a SyCyR consisting of the extracellular $G_{VHH}$ fused to the transmembrane and intracellular domains of the IL-23R ($G_{VHH}$-IL-23R; (SyCyR(IL-23R)) (Fig. 5a and Supplementary Fig. 1). Interestingly, 3xGFP and 2xGFP-mCherry fusion proteins induced proliferation of Ba/F3-SyCyR(IL-23R) cells (Fig. 5b), whereas single GFP and mCherry or the heterodimeric GFP-mCherry fusion protein did not (Fig. 5b). Half-maximal proliferation was achieved at about 10–20 ng/ml 3xGFP (86 kDa) (Fig. 5c), which was only slightly lower as compared to Ba/F3-IL-12Rβ1-IL-23R cells stimulated with HIL-23 (56 kDa; 5–10 ng/ml). Thus, the slightly reduced activity cannot be explained through differences in the molar concentrations of the molecules. As expected, 3xGFP and 2xGFP-mCherry fusion proteins but not single GFP induced JAK, STAT3, pERK1/2, and AKT phosphorylation in Ba/F3-SyCyR(IL-23R) cells (Fig. 5d, e). Also the kinetics of pSTAT3 activation and SOCS3 expression of HIL-23 and 3xGFP were comparable (Figs. 5f and 2f), suggesting that also IL-23R homodimers were not negatively regulated by SOCS3. Surprisingly, only 35 genes were up- or downregulated by at least 1.5-fold after stimulation of Ba/F3-SyCyR(IL-23R) with 3xGFP. However, 34 out of 35 transcripts were also found in the GFP-mCherry-group (Ba/F3-SyCyR(IL-23/2 A)) (Fig. 6a). Among the 35 regulated genes are typical IL-23 target genes, including PIM,

SOCS3, and OSM (Fig. 6b, c). Although, a reduced number of genes was triggered by homodimeric IL-23R signaling when compared to IL-12Rβ1-IL-23R heterodimer stimulation, similar signaling pathways were affected (Figs. 6c, 3d and Supplementary Figs. 5–8). In conclusion, homotypic activation of SyCyR(IL-23R) also phenocopied IL-23 signaling in terms of signal transduction pathways and kinetics, but resulted in overall reduced induction of gene expression as compared to SyCyR(IL-23/2A).

**Heterotrimeric SyCyRs induce STAT3 trans-phosphorylation.** During trans-phosphorylation a kinase-active receptor is able to trans-phosphorylate a kinase-negative mutant receptor. Since 2xGFP-mCherry was able to induce functional hetero- and homodimerization of SyCyRs, we wondered whether 2xGFP-mCherry can induce trans-phosphorylation of STAT3 via synthetic trimeric receptor complexes. Hence, a C-terminally truncated IL-23R (Δ503), lacking the canonical STAT-binding motifs but retained JAK, ERK, and AKT activity (IL-23R-ΔSTAT) was selected and fused with $G_{VHH}$ (Fig. 7a and Supplementary Fig. 9a). Janus kinases interact with peptide motifs within the IL-23R localized between amino acid 403–479 and complete or partial deletion results in disabled JAK activity[18]. Accordingly, we created three deletion variants of the IL-23R intracellular domain with disabled JAK activity, ΔJAK-A (Δ403–417), ΔJAK-B (Δ455–479), and ΔJAK-C (Δ403–479)[20] fused to the mCherry-nanobody (Fig. 7a and Supplementary Fig. 9a). Cell surface expression of these SyCyRs in Ba/F3-gp130 cells was verified by flow cytometry (Supplementary Fig. 9b). As expected, stimulation of $G_{VHH}$-IL-23R-ΔSTAT in Ba/F3-gp130 cells with 2xGFP-mCherry resulted in JAK and ERK phosphorylation but defective STAT3 activation (Fig. 7b, c). Consequently, 3xGFP-induced proliferation of Ba/F3-$G_{VHH}$-IL-23R-ΔSTAT cells was drastically reduced as compared to Ba/F3-SyCyR(IL-23R) cells (Fig. 7d). Interestingly, only the assembly of a 2xGFP-mCherry-induced trimeric complex consisting of two $G_{VHH}$-IL-23R-ΔSTAT receptors and one $C_{VHH}$-IL-23R-ΔJAK receptor resulted in increased STAT3 trans-phosphorylation and cellular proliferation (Fig. 7c, e). Dimerization of $G_{VHH}$-IL-23R-ΔSTAT with all $C_{VHH}$-ΔJAK receptors by GFP-mCherry did not induce STAT activation, demonstrating that two biologically active JAKs in $G_{VHH}$-IL-23R-ΔSTAT were needed for STAT3 trans-phosphorylation. Of note, also stimulation with a 2xmCherry fusion protein and formation of dimeric $C_{VHH}$-IL-23R-ΔJAK did not result in STAT3 phosphorylation, whereas homodimers of $C_{VHH}$-IL-23R were biologically active (Supplementary Fig. 10), demonstrating that the $C_{VHH}$-IL-23R-ΔJAK variants were not biologically active as dimers.

**Fig. 1** SyCyRs for IL-23 (SyCyR(IL-23/2A)) simulate IL-23-induced signal transduction. **a** Schematic illustration of IL-23 SyCyR and GFP-mCherry fusion protein. $G_{VHH}$-IL-12Rβ1: $G_{VHH}$ fused to 15 aa of the ECD, TMD, and ICD of IL-12Rβ1. $C_{VHH}$-IL-23R: $C_{VHH}$ fused to 17 aa of ECD, TMD, and ICD of IL-23R. **b** Proliferation of Ba/F3-gp130 cell lines with HIL-6 and GFP:mCherry cultured with synthetic ligands (6.25 ng/ml) or HIL-6 (10 ng/ml). Stimulation with mCherry was made with the same volume as with GFP (0.25%). One representative experiment out of three is shown. **c** Proliferation of Ba/F3-SyCyR(IL-23/2A) and Ba/F3-gp130 cells with GFP-mCherry and Ba/F3-IL-12Rβ1-IL-23R cells with HIL-23 cultured in the presence of ligands (0.1 to 1600 ng/ml). One representative experiment out of five is shown. **d** Proliferation of Ba/F3-SyCyR(IL-23/2A) and Ba/F3-gp130 cells with GFP-mCherry fusion protein in the presence of $G_{VHH}$–$C_{VHH}$. Cells were cultured in the presence of GFP-mCherry (GC; 10 ng/ml) and increasing concentrations of $G_{VHH}$–$C_{VHH}$ (0.1–100 ng/ml). One representative experiment out of four is shown. Results in **b**, **c**, and **d** are mean ± s.d. of three replicates. **e** JAK activation in Ba/F3-gp130, Ba/F3-IL-12Rβ1-IL-23R, and Ba/F3-SyCyR(IL-23/2A) cells stimulated with 100 ng/ml of indicated synthetic ligands or HIL-6 (100 ng/ml) for 20 min. Equal amounts of total protein (50 μg/lane) were analyzed using specific antibodies for phospho-JAK2 and JAK2. Western blot data show one representative experiment out of three. **f** STAT3, ERK1/2, and AKT activation in Ba/F3-SyCyR(IL-23/2A) and Ba/F3-gp130 cells treated with 100 ng/ml of the indicated ligands or HIL-6 (10 ng/ml) for 30 min. Stimulation with mCherry was made with the same volume as with GFP (2%). Equal amounts of total protein (50 μg/lane) were analyzed using specific antibodies for phospho-STAT3/ERK1/2/AKT and STAT3/ERK1/2/AKT. Western blot data show one representative experiment out of three. **g** STAT3 activation in U4C cells expressing SyCyR receptors were stimulated with 100 ng/ml of the indicated synthetic ligands or HIL-6 (10 ng/ml) for 60 min. Equal amounts of total protein (25 μg/lane for HIL-6 and 50 μg/lane for other ligands) were analyzed using specific antibodies for phospho-STAT3 and STAT3. Western blot data show one representative experiment out of two

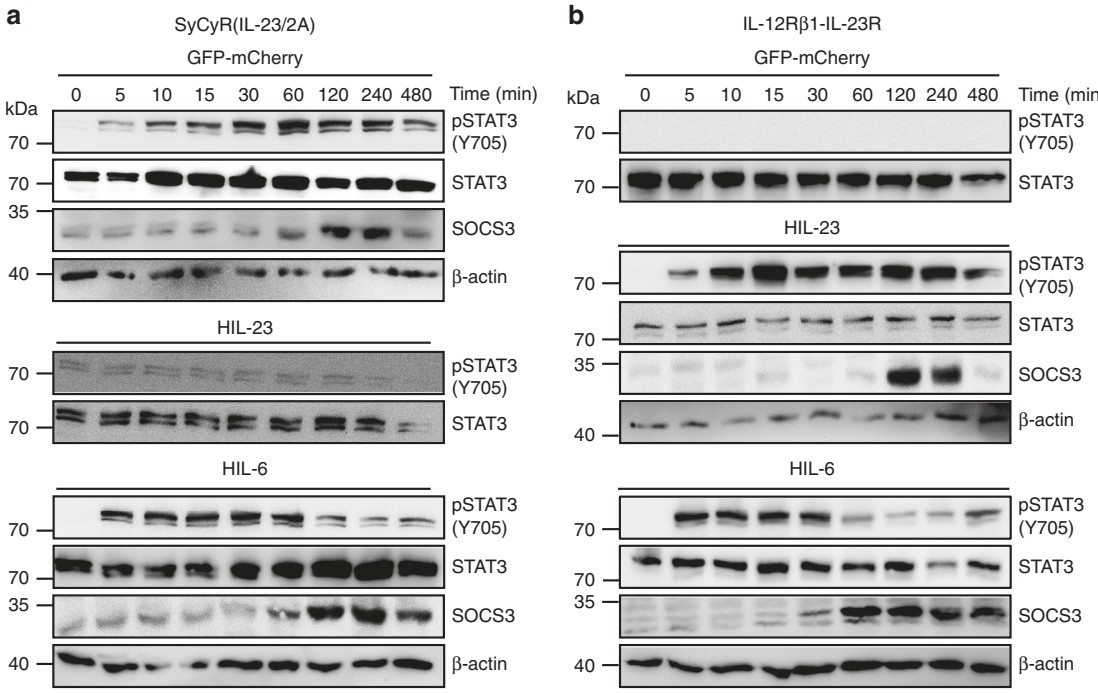

**Fig. 2** Synthetic cytokine receptors for IL-23 (SyCyR(IL-23/2A)) simulate kinetics of IL-23-induced STAT3 activation and SOCS expression in transduced Ba/F3-gp130 cells. **a** Ba/F3-SyCyR(IL-23/2A) cells were washed three times, starved, and stimulated with GFP-mCherry (100 ng/ml), HIL-23 (25 ng/ml), and HIL-6 (25 ng/ml) for 0–480 min. Stimulation with HIL-23 and HIL-6 were used as control. Cellular lysates were prepared, and equal amounts of total protein (50 µg/lane) were loaded on SDS gels, followed by immunoblotting using specific antibodies for phospho-STAT3 and STAT3, SOCS3, and β-actin. Western blot data show one representative experiment out of three. **b** Ba/F3-IL-12Rβ1-IL-23R cells were washed three times, starved, and stimulated with HIL-23 (25 ng/ml) for 0–480 min. Stimulation with GFP-mCherry (100 ng/ml) and HIL-6 (25 ng/ml) were used as control. Cellular lysates were prepared, and equal amounts of total protein (50 µg/lane) were loaded on SDS gels, followed by immunoblotting using specific antibodies for phospho-STAT3 and STAT3, SOCS3, and β-actin. Western blot data show one representative experiment out of three

Cytokine receptors can be classified into high-affinity binders with longer ICDs and lower-affinity binders with shorter ICDs. Interestingly, high-affinity receptors with STAT-binding sites often heterodimerize with JAK1 or JAK2, whereas low-affinity receptors often pair with TYK2 or JAK3, and minimally contribute to STAT recruitment and activation[21, 22]. Combining the short ICD- and TYK-binding-receptor $G_{VHH}$-IL-12Rβ1 with $C_{VHH}$-IL-23R-ΔJAK-B, in Ba/F3-gp130 cell lines did not result in trans-phosphorylation and proliferation (Fig. 8b, c and Supplementary Fig. 11), demonstrating that this combination fails to induce receptor activation.

Finally, we asked if the SyCyRs $C_{VHH}$-IL-23R-ΔJAK-B might be trans-phosphorylated by $G_{VHH}$-gp130 (SyCyR(IL-6)). We generated a C-terminally truncated gp130 (Δ758) lacking both the STAT and ERK/AKT activation motifs[23] fused to the $G_{VHH}$ ($G_{VHH}$-gp130-ΔSTAT) (Fig. 8d and Supplementary Fig. 12). Ba/F3-gp130 cells expressing $G_{VHH}$-gp130-ΔSTAT displayed no STAT3 phosphorylation after stimulation with 2xGFP-mCherry and 3xGFP (Fig. 8e). In analogy to the IL-23R-ΔSTAT-SyCyRs, the combined activation of $G_{VHH}$-gp130-ΔSTAT with $C_{VHH}$-IL-23R-ΔJAK-B resulted in STAT3 trans-phosphorylation and cellular proliferation (Fig. 8e, f).

## Discussion

Here, we describe the development of a synthetic cytokine receptor system based on nanobodies directed against GFP and mCherry fused to truncated cytokine receptors. Nanobodies are versatile tools widely used in molecular biology, exhibiting high-affinity and antigen specificity. We chose nanobodies against GFP and mCherry because these fluorescent proteins are non-toxic to

mammalian cells, will not cause unspecific binding to endogenous receptors and are, therefore, considered as side-effect/background-free[24, 25]. As receptor system, we used heterodimeric and homodimeric cytokine receptor compositions exemplified by IL-23 and IL-6 receptor signaling complexes. IL-23 signals via a heterodimeric receptor complex consisting of IL-12Rβ1 and IL-23R, whereas IL-6 signals via the non-signal-transducing IL-6R and the signal-transducing homodimer of gp130[19]. Both receptor complexes induce signals via receptor-associated Janus kinases that activate STAT, ERK, and AKT pathways[26, 27]. JAKs are constitutive but non-covalently associated with class I and II cytokine receptors, which upon cytokine binding bring together two JAKs to create an active signaling complex. JAK interact with receptor peptide motifs, which are present in the intracellular domain of cytokine receptors. During receptor activation, JAKs switch into the "on"-status by reciprocal phosphorylation and subsequent phosphorylation of receptor-tyrosines and signaling molecules such as STAT3[28].

The synthetic receptor complex mimicking IL-23-signaling was activated by a heterodimeric synthetic GFP-mCherry ligand but not by single GFP or mCherry or multimeric GFP fusion proteins, whereas the synthetic receptor complex simulating IL-6-signaling was specifically activated by homodimeric synthetic GFP-ligands. Importantly, GFP-mCherry and 3xGFP fusion proteins did not activate cellular responses in cells lacking synthetic cytokine receptors.

A recent report showed that not only the intracellular domains determine the signaling strength but also the mode of extracellular receptor complex assembly[29]. Specifically, a point mutation in EPO was shown to change EPO receptor dimerization, which resulted in reduced STAT1 and STAT3 phosphorylation

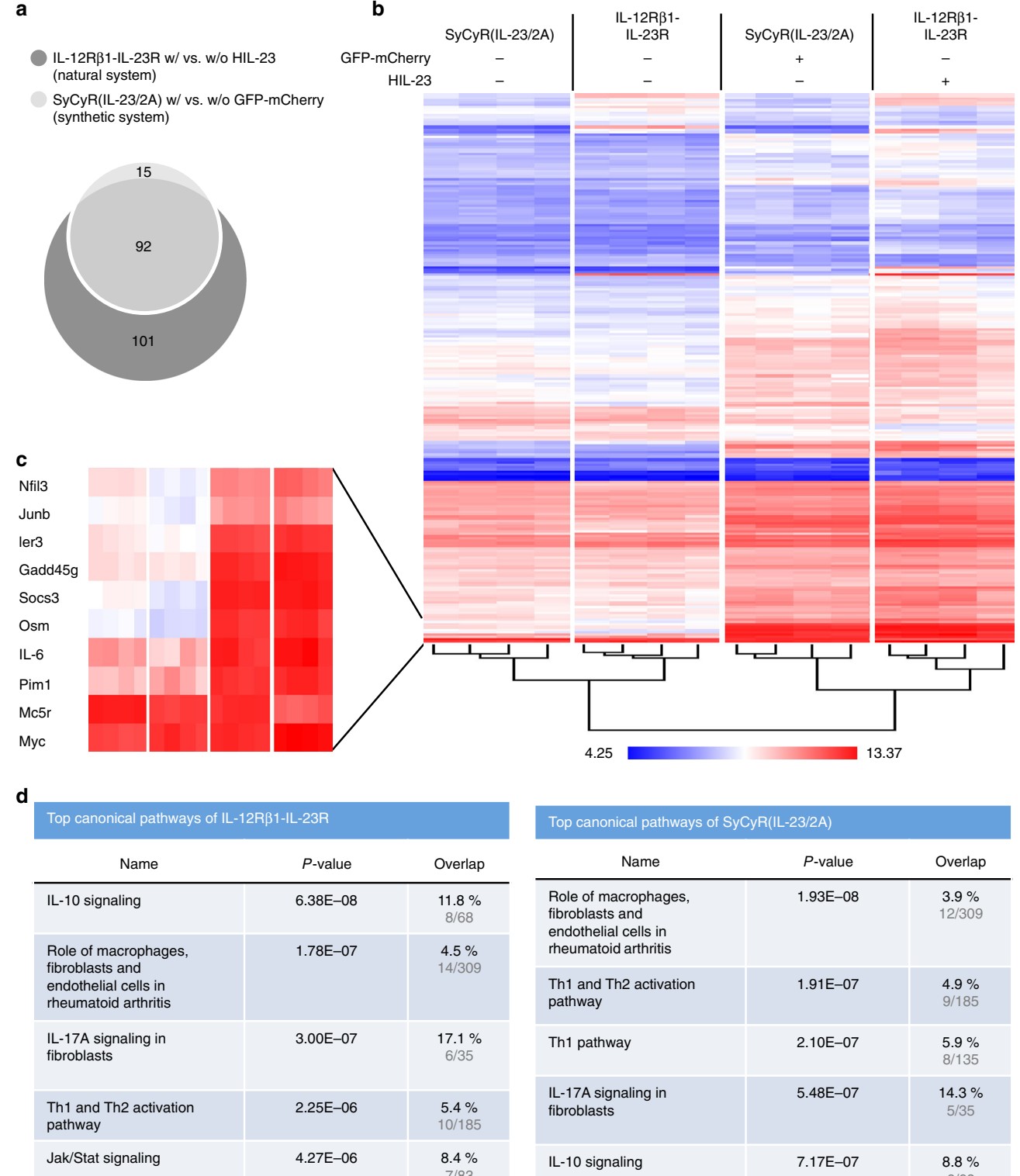

**Fig. 3** Microarray analysis of Ba/F3-SyCyR(IL-23/2 A) cells. **a** Venn-diagram of at least 1.5-fold (*p*-value ≤ 0.01) up- or downregulated mRNAs of Ba/F3-SyCyR(IL-23/2A) cells stimulated with 100 ng/ml GFP-mCherry (light gray) and Ba/F3-IL-12Rβ1-IL-23R cells stimulated with 100 ng/ml HIL-23 (dark gray) for 60 min. Microarray analysis was performed using samples of four independent biological replicates. **b** Heat map comparing mRNA levels of Ba/F3-SyCyR(IL-23/2A) and Ba/F3-IL-12Rβ1-IL-23R stimulated and unstimulated as described in **a**. **c** Higher magnification of selected mRNAs from **b**. **d** Ingenuity pathway analysis (IPA) revealed the top five canonical pathways of Ba/F3-SyCyR(IL-23/2A) vs. Ba/F3-IL-12Rβ1-IL-23R

but did not affect STAT5 activation[29]. This implies, that replacing the extracellular part of a cytokine receptor by another binding domain, such as nanobodies might influence signaling strength and kinetics, which ultimately lead to an altered intracellular response of the chimeric receptor. To exclude such effects for our synthetic cytokine receptors, apart from general analysis of typical signal transduction pathways (JAK/STAT, ERK, AKT), we

verified that the time-dependent activation profile of IL-23 and IL-6 is identical with those of synthetic ligands. These findings were supported by transcriptome comparison of Ba/F3-IL-23R-IL-12Rβ1 and Ba/F3-SyCyR(IL-23/2A) cells, activated by HIL-23 and GFP-mCherry, respectively, in which almost all regulated genes for both cytokines were identical. Even though more regulated genes were detected after HIL-23 stimulation as

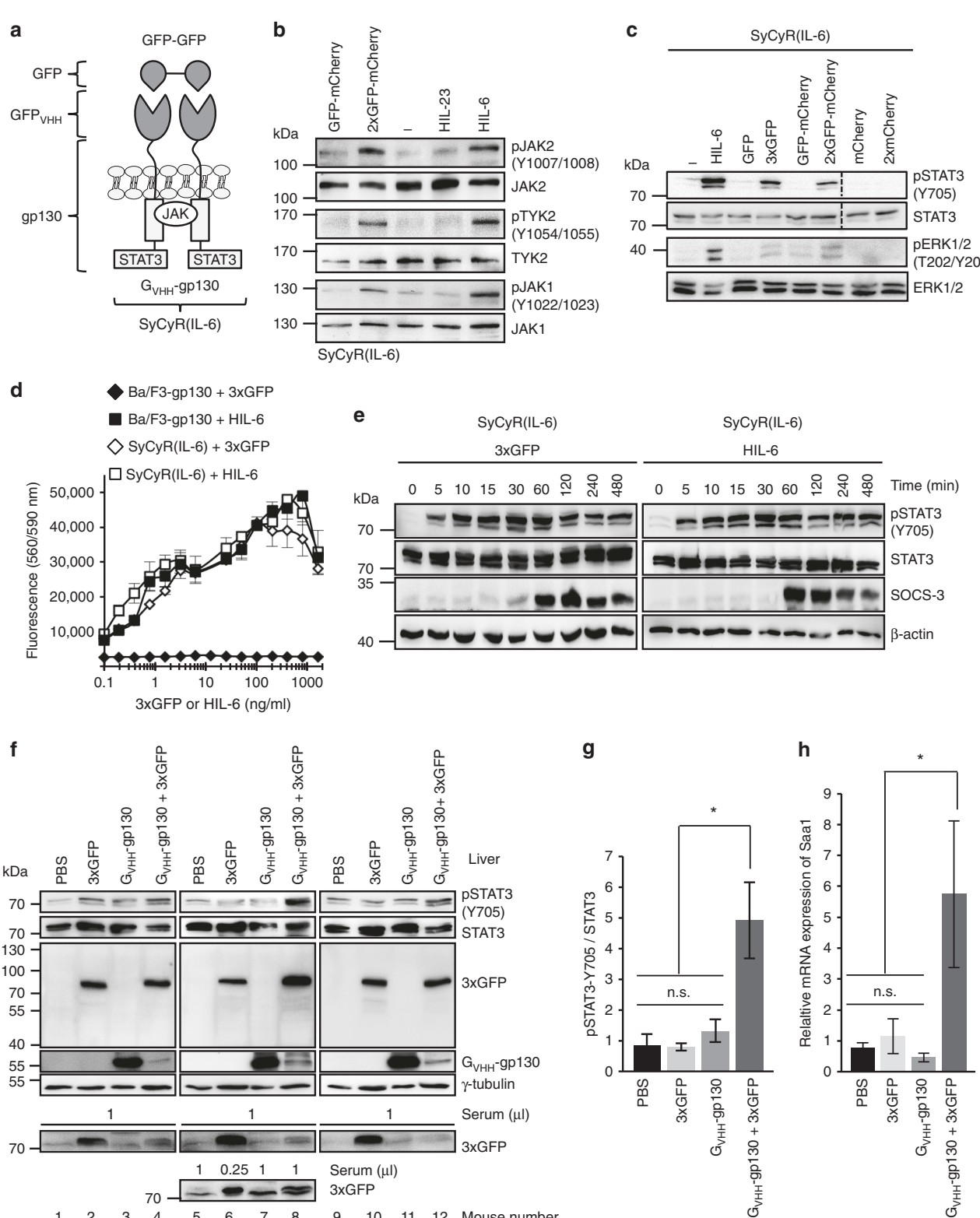

compared to GFP-mCherry stimulation, this difference was within expected fluctuations when using cell lines, which have been transduced with different receptor complexes and have been independently selected and cultivated for several weeks. Importantly, pathway analysis of the natural and synthetic IL-23 receptor complexes highlighted that naturally and synthetically induced signal transduction was basically identical. To the best of our knowledge, this is the first study using chimeric receptor complexes, which included a detailed analysis of signal transduction pathways and expression profiles. Importantly, the synthetic receptors appear to be active in vivo, since we demonstrated the activation of $G_{VHH}$-gp130 by 3xGFP in the liver of mice after hydrodynamic injection.

So far, no signaling role of IL-12Rβ1 in the receptor complex apart from activation of a Janus kinase has been assigned[20]. Consistently, using our synthetic receptors, we induced IL-23R homodimeric receptor complexes, as we have recently suggested[30, 31]. Although the general activation of signaling pathways appeared to be similar to IL-23 signaling, gene-array analysis revealed a reduced number of regulated genes when compared to heterodimeric signaling. This phenomenon could be caused by the slightly reduced proliferation observed following SyCyR(IL-23R) signaling when compared to natural IL-23R/IL-12Rβ1 complex. Moreover, reduced affinity or biochemical features of the synthetic ligand may affect SyCyR signaling. This effect might also contribute to the observation, that the natural and synthetic IL-23 receptor activation was different in terms of the absolute number of regulated genes.

The modular nature of the synthetic ligands, with one receptor binding site per GFP or mCherry allows an exact composition of the receptor stoichiometry, which clearly will be interesting for many if not all other cytokine receptors. Moreover, this system will enable the combinatory assembly of uncommon receptor combinations with desirable signaling potentials and capacities. The number of recruited synthetic receptors is only limited by the maximal number of ligands connected in one GFP:mCherry fusion protein or by alternative GFP/mCherry multimerization strategies[32, 33].

Two recent reports describe surrogate dimeric ligands, which specifically bind and activate natural cytokine receptors supporting our hypothesis that receptor activation might be generally possible by dimeric ligands[34, 35]. For IL-6 and IL-23 signaling, we used homo- and heterodimeric GFP:mCherry fusion proteins, but we were also able to generate homo- and heterotrimeric GFP:mCherry variants. Using these synthetic ligands, we analyzed, if biologically active trimeric receptor complexes could also be functionally assembled among the cytokine receptor family with

associated kinases. Tyrosine-receptors and receptors with associated kinases are typically active as dimers[36] to juxtapose and subsequently activate at least two receptor kinases. This implies that these receptor systems naturally did not require a third receptor. To generate trimeric receptor complexes for IL-6 and IL-23-simulations, we deleted the STAT3-binding motifs in the synthetic $G_{VHH}$-IL-23 and $G_{VHH}$-gp130 receptors and combined these receptors with JAK-deficient receptors containing STAT-binding motifs fused to $C_{VHH}$. We showed that in these trimeric receptor combinations, STAT3 activation is mediated by trans-phosphorylation. The formation of a trimeric receptor complex with two JAK-proficient but STAT-deficient receptors and one STAT-binding motif receptor by GFP-GFP-mCherry (2xGFP-mCherry) resulted in STAT3 trans-phosphorylation. Assembly of one JAK-proficient receptor with one STAT-binding motif receptor by a GFP-mCherry fusion did, however, not lead to trans-phosphorylation, confirming that one Janus kinase is not sufficient for receptor activation. Of note, trans-phosphorylation was thus far only described for tyrosine-kinase-receptors of the PDGF and EGF family[37–41], in which a kinase-active receptor was able to trans-phosphorylate a second kinase-inactive mutant receptor after receptor dimerization. In these cases, the kinase-negative mutant receptor was able to activate the functional kinase of the other receptor. Here, we describe for the first receptor-chain trans-phosphorylation for cytokine receptors with associated Janus kinases.

In summary, the synthetic cytokine receptor system allows tailor-made activation and analysis of cytokine signaling by recruitment of defined numbers and compositions of receptor chains. Receptor assembly is determined by the number and sequence of GFP-mCherry units in the ligand fusion proteins. This system simulates signal transduction without relevant background activation that has been described previously with chimeric receptor systems[6]. The lack of toxicity of fluorescent proteins in vitro and in vivo allows a widespread area of potential applications for studying cell-type specific receptor activation by synthetic ligand application in transgenic mice. Importantly, our system is easily on/off-switchable, because signal activation can be rapidly inhibited by application of soluble nanobody-fusion proteins directed against the synthetic GFP:mCherry ligands and will open up therapeutic regimes involving non-physiological targets during immunotherapy.

## Methods

**Cells and reagents.** CHO-K1 (ACC-110) cells were from Leibniz Institute DSMZ-German Collection of Microorganisms and Cell Cultures (Braunschweig, Germany). U4C cells were kindly provided by Heike Hermanns (University Würzburg,

---

**Fig. 4** SyCyRs for IL-6 (SyCyR(IL-6)) simulate IL-6/IL-11-induced signal transduction. **a** Schematic illustration of IL-6 SyCyR and GFP-GFP. $G_{VHH}$-gp130: $G_{VHH}$ fused to 13 aa of the ECD, TMD, and ICD of gp130. **b** JAK and TYK activation in Ba/F3-SyCyR(IL-6) cells stimulated with supernatants containing 100 ng/ml of indicated synthetic ligands or HIL-6 (100 ng/ml) for 20 min. Equal amounts of total protein (50 μg/lane) were analyzed for phospho-JAK2/TYK2/JAK1 and JAK2/TYK2/JAK1. Western blot data show one representative experiment out of three. **c** STAT3 and ERK1/2 activation in Ba/F3-SyCyR (IL-6) cells stimulated with 4% conditioned supernatant containing mCherry, 2xmCherry or conditioned supernatants containing 100 ng/ml of indicated synthetic ligands for 60 min. Stimulation with HIL-6 (10 ng/ml) for 15 min was used as control. Equal amounts of total protein (50 μg/lane) were analyzed for phospho-STAT3/ERK1/2 and STAT3/ERK1/2. One representative experiment out of two is shown. **d** Proliferation of Ba/F3-SyCyR(IL-6) and Ba/F3-gp130 cells with 3xGFP and HIL-6 cultured in the presence of ligands (0.1 to 1600 ng/ml). One representative experiment out of five is shown. Results are mean ± s.d. of three replicates. **e** Ba/F3-SyCyR(IL-6) cells were stimulated with 3xGFP (100 ng/ml) and HIL-6 (25 ng/ml) for 0-480 min. Total protein (50 μg/lane) were analyzed for phospho-STAT3 and STAT3, SOCS3, and β-actin. Western blot data show one representative experiment out of three. **f** Western blot of mouse liver lysates and mouse serum prepared 24 h following hydrodynamic transfection of PBS, pcDNA3.1-3xGFP (1.28 μg) and/or pcDNA3.1-$G_{VHH}$-gp130 (2.3 μg) plasmid DNA. Total proteins (50 μg/lane) were analyzed for phospho-STAT3, STAT3, GFP, and γ-tubulin. Serum volumes from 0.25–1 μl were analyzed for GFP; $n = 3$ animals/group. **g** Ratio of relative density of pSTAT3 and STAT3 liver expression as determined in **f**; Results are mean ± s.e.m. of three animals/group. Significance of difference (one-way ANOVA): *$p < 0.05$. **h** qRT-PCR of Saa1 mRNA 24 h after hydrodynamic transfection of 1.28 μg pcDNA3.1-3xGFP and/or 2.3 μg pcDNA3.1-$G_{VHH}$-gp130 plasmid and normalized using the housekeeper mRNA Gapdh and the ΔΔCT method; Results are mean ± s.e.m. of six animals/group. Significance of difference (one-way ANOVA): *$p < 0.05$

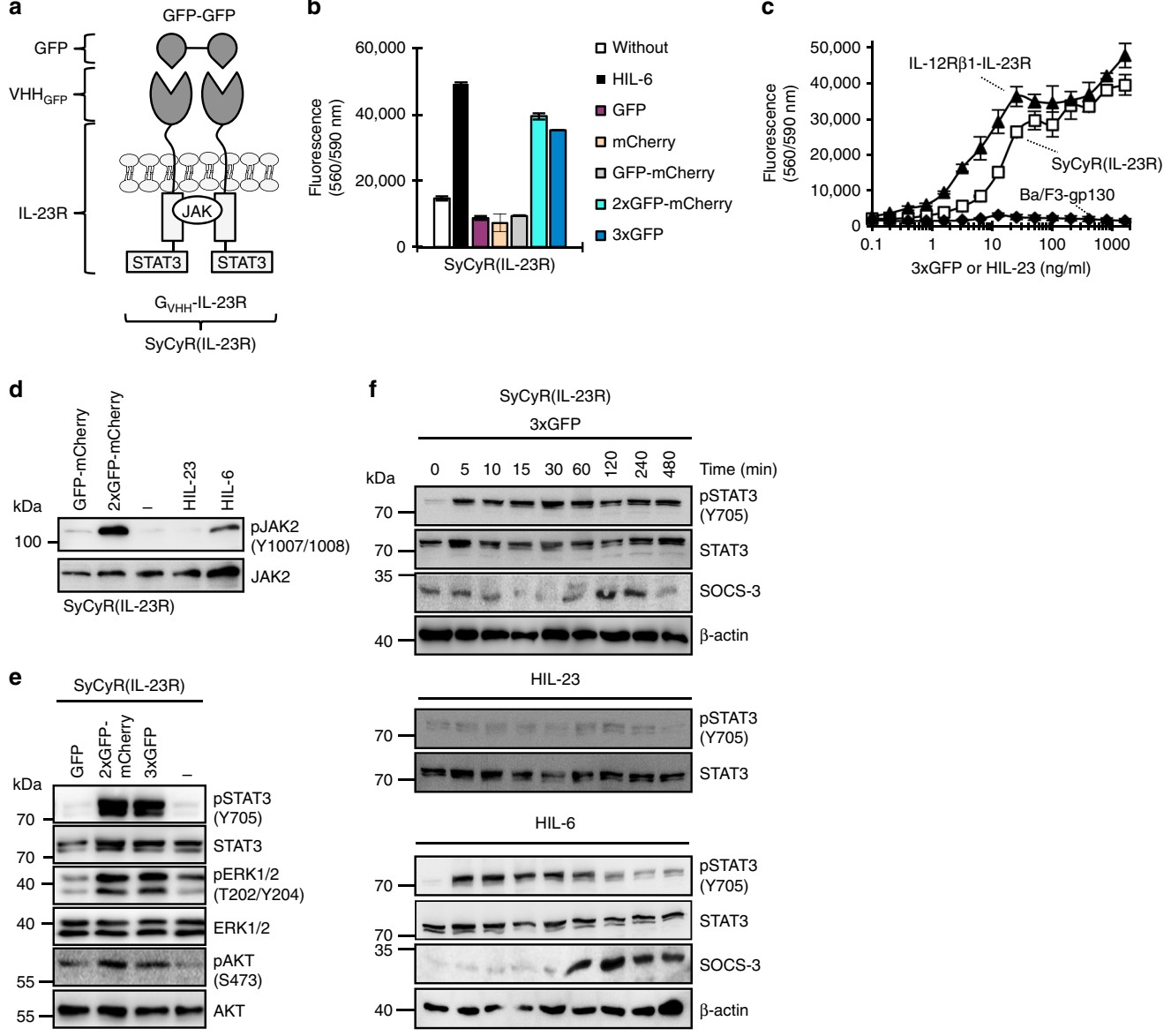

**Fig. 5** SyCyRs for homodimeric IL-23R (SyCyR(IL-23R)) induced signal transduction. **a** Schematic illustration of homodimeric IL-23R SyCyR and GFP-GFP fusion protein. G$_{VHH}$-IL-23R: G$_{VHH}$ fused to 17 aa of the ECD, TMD, and ICD of IL-23R. **b** Proliferation of Ba/F3-SyCyR(IL-23R) cells with HIL-6 and GFP: mCherry fusion proteins cultured in the presence of the indicated synthetic ligands (6.25 ng/ml) or HIL-6 (10 ng/ml). Stimulation with mCherry was made with the same volume as with GFP (0.25%). One representative experiment out of three is shown. Results are mean ± s.d. of three replicates. **c** Proliferation of Ba/F3-SyCyR(IL-23R) with HIL-23, Ba/F3-IL-12Rβ1-IL-23R, and Ba/F3-gp130 cells with 3xGFP and HIL-23 cultured in the presence of ligands (0.1 to 1600 ng/ml). One representative experiment out of six is shown. Results are mean ± s.d. of three replicates. **d** JAK activation in Ba/F3-SyCyR(IL-23R) cells. Cells were stimulated with conditioned supernatants containing 100 ng/ml of the indicated synthetic ligands or HIL-6 (100 ng/ml) for 20 min. Equal amounts of total protein (50 μg/lane) were analyzed for phospho-JAK2 and JAK2. Western blot data show one representative experiment out of three. **e** STAT3, ERK1/2, and AKT activation in Ba/F3-SyCyR(IL-23R) cells stimulated with 100 ng/ml of the indicated ligands for 30 min. Stimulation with mCherry was made with the same volume as with GFP (2%). Equal amounts of total protein (50 μg/lane) were analyzed for phospho-STAT3/ERK1/2/AKT and STAT3/ERK1/2/AKT. Western blot data show one representative experiment out of three. **f** Ba/F3-SyCyR(IL-23R) cells were stimulated with 3xGFP (100 ng/ml), HIL-23 (25 ng/ml), and HIL-6 (25 ng/ml) for 0–480 min. Equal amounts of total protein (50 μg/lane) were analyzed for phospho-STAT3, STAT3, SOCS3, and β-actin. Western blot data show one representative experiment out of three

Germany). Murine Ba/F3-gp130 cells transduced with human gp130 were provided by Immunex (Seattle, WA, USA)[42]. The packaging cell line Phoenix-Eco was obtained from Ursula Klingmüller (DKFZ, Heidelberg, Germany)[43]. Ba/F3-gp130 cells with murine IL-12Rβ1 and murine IL-23R have been described previously[18]. All cell lines were grown in Dulbecco's modified Eagle medium (DMEM) high-glucose culture medium (GIBCO®, Life Technologies, Darmstadt, Germany) supplemented with 10% fetal calf serum (GIBCO®, Life Technologies, Darmstadt, Germany), 60 mg/l penicillin and 100 mg/l streptomycin (Genaxxon Bioscience GmbH, Ulm, Germany) at 37 °C with 5% $CO_2$ in a water saturated atmosphere. Ba/F3-gp130 cells were maintained in the presence of Hyper-IL-6 (HIL-6), a fusion

protein of IL-6 and the soluble IL-6R, which mimics IL-6 trans-signaling[44]. Either recombinant protein (10 ng/ml) or 0.2% (10 ng/ml) of conditioned cell culture medium from a stable CHO-K1 clone secreting Hyper-IL-6 (stock solution approx. 5 μg/ml as determined by ELISA) were used to supplement the growth medium. Ba/F3-IL-12Rβ1-IL-23R cells expressing murine IL-23R and murine IL-12Rβ1 were stimulated with 0.2% (10 ng/ml) of conditioned cell culture medium from a stable CHO-K1 clone secreting murine Hyper-IL-23 (HIL-23) in a concentration of approx. 5 μg/ml, as determined by ELISA[18]. Phospho-STAT3 (Tyr705) (D3A7) (1:1000, cat. #9145), STAT3 (124H6) (1:1000, cat. #9139), phospho-p44/42 MAPK (ERK1/2) (Thr-202/Tyr-204) (D13.14.4E) (1:1000, cat. #4370), p44/42 MAPK

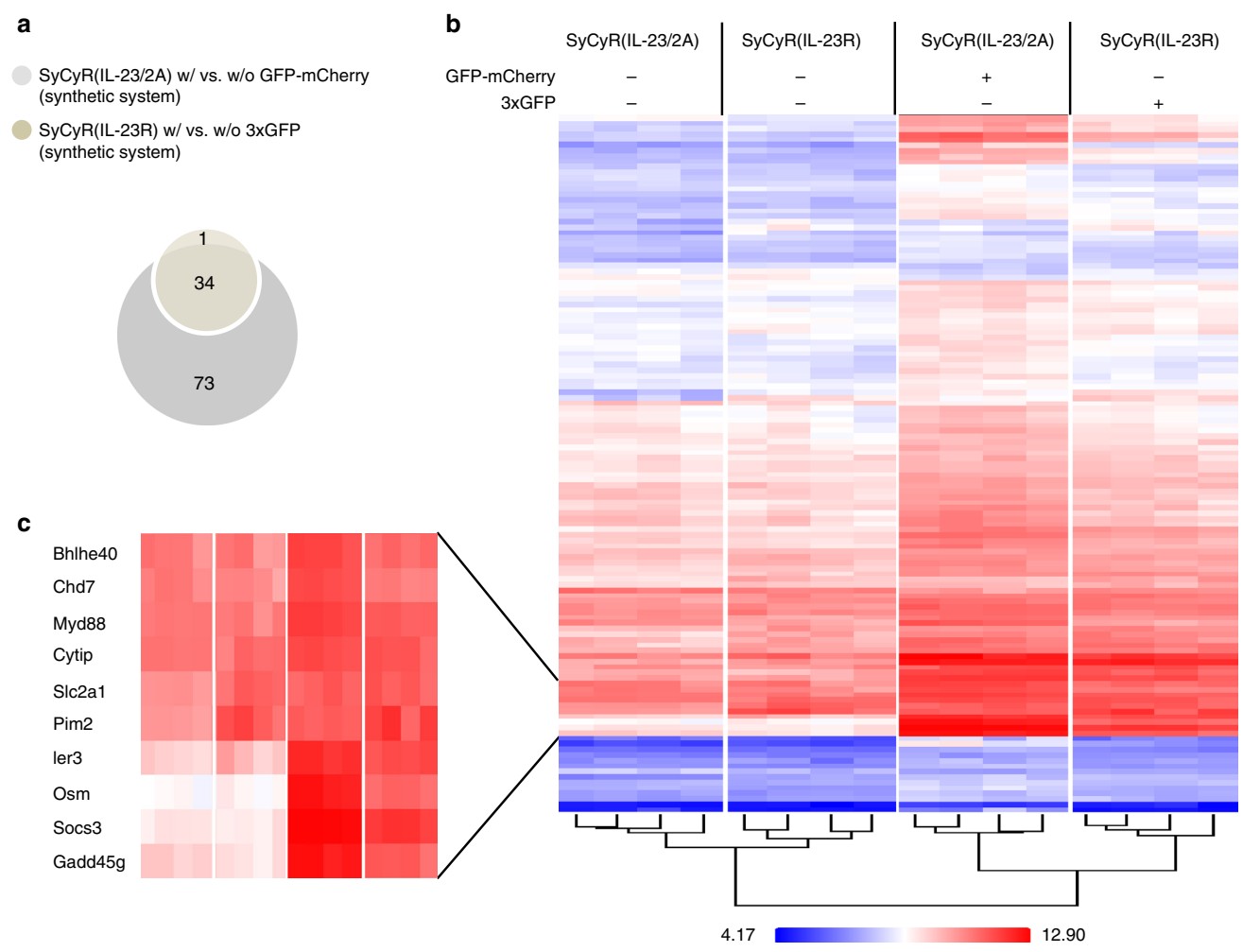

**Fig. 6** Microarray analysis of Ba/F3-SyCyR(IL-23R) cells. **a** Venn-diagram of at least 1.5-fold (*p*-value ≤ 0.01) up- or downregulated mRNAs of Ba/F3-SyCyR(IL-23R) cells stimulated with 100 ng/ml 3xGFP (yellow) and Ba/F3-SyCyR(IL-23/2A) cells stimulated with 100 ng/ml GFP-mCherry (light gray) for 60 min. Microarray analysis was performed using samples of four independent biological replicates. **b** Heat map comparing mRNA levels of Ba/F3-SyCyR (IL-23R) and Ba/F3-SyCyR(IL-23/2A) stimulated and unstimulated as described in **a**. **c** Higher magnification of selected mRNAs from **b**. **d** Ingenuity pathway analysis (IPA) revealed the top five canonical pathways of Ba/F3-SyCyR(IL-23/2A) vs. Ba/F3-SyCyR(IL-23R)

(ERK1/2) (1:1000, cat. #9102), phospho-AKT (Ser473) (D9E) (1:000, cat. #4060), AKT (1:1000, cat. #9272S), phospho-JAK1 (Tyr1022/1023) (1:000, cat. #3331), JAK1 (6G4) (1:1000, cat. #3344S), phospho-JAK2 (Tyr1007/1008) (1:1000, cat. #3771), JAK2 (D2E12) (1:000, cat. #3230), phospho-TYK2 (Tyr1054/1055) (1:000, cat. #9321), TYK2 (1:1000, cat. #9312), GFP (4B10) (1:000, cat. #2955) and myc (71D10), (1:1000 for western blotting and 1.2 μg for flow cytometry, cat. #2278) monoclonal antibodies (mAbs) were obtained from Cell Signaling Technology (Frankfurt, Germany). mCherry (1:1000, cat. 31451) was obtained from Thermo Fisher Scientific (Waltham, MA, USA). Flag (DYKDDDDK) (1:1000 for western blotting and 1:100 for flow cytometry, cat. F7425) and γ-tubulin (1:5000, cat. T5326) mAbs were obtained from Sigma Aldrich (Munich, Germany). Human CIS3/SOCS3 (C204) mAb (1:500, cat. JP18391) was obtained from Immuno-Biological Laboratories Co. Ltd. (Fujioka, Japan). β-actin (C4) mAb (1:500, cat. sc-47778) was obtained from Santa Cruz Biotechnology (Dallas, USA). Peroxidase-conjugated secondary mAbs (1:2500, cat. 31462, cat. 31451) were obtained from Pierce (Thermo Fisher Scientific, Waltham, MA, USA). Alexa Fluor 488 conjugated Fab goat anti-rabbit IgG (1:100, cat. A11070) was obtained from Thermo Fisher Scientific (Waltham, MA, USA). PageRuler Prestained Protein Ladder (cat. #26616) was obtained from Thermo Fisher Scientific (Waltham, MA, USA).

**Construction SyCyRs and synthetic ligands.** pcDNA3.1-$G_{VHH}$-IL-23R expression vector was generated by fusion of coding sequences for human IL-11R signal peptide (Q14626, aa 1-24) followed by sequences for myc tag (EQKLISEEDL), GFP-nanobody ($G_{VHH}$)[15] and murine IL-23R (Q8K4B4) comprising amino acids A358 to K644 representing 17 aa of the extracellular domain, the transmembrane domain and the cytoplasmic part of the receptor. The cDNA coding for $G_{VHH}$-IL-12Rβ1 was generated by insertion of cDNA coding for murine IL-12Rβ1 (Q60837, aa A551-A738, representing 15 aa of the extracellular domain, the transmembrane domain and the cytoplasmic part of the receptor), which was amplified by PCR from p409-IL-12Rβ1[18], into expression vector pcDNA3.1-$G_{VHH}$-IL-23R, where the coding sequence for IL-23R was removed. pcDNA3.1-$C_{VHH}$-IL-23R expression vector was generated by fusion of coding sequences for human IL-11R signal peptide (Q14626, aa 1-24) followed by sequences for Flag-tag (DYKDDDDK), mCherry-nanobody ($C_{VHH}$)[14] and murine IL-23R (Q8K4B4) comprising amino acids A358 to K644 (representing 17 aa of the extracellular domain, the transmembrane domain and the cytoplasmic part of the receptor). To combine cDNAs coding for $C_{VHH}$-IL-23R and $G_{VHH}$-IL-12Rβ1 in one open reading frame, cDNAs coding for both SyCyRs were amplified by PCR and cloned into pMK-FUSIO coding for the self-processing 2A-peptide[4, 45]. The cDNA coding for IL-23R-

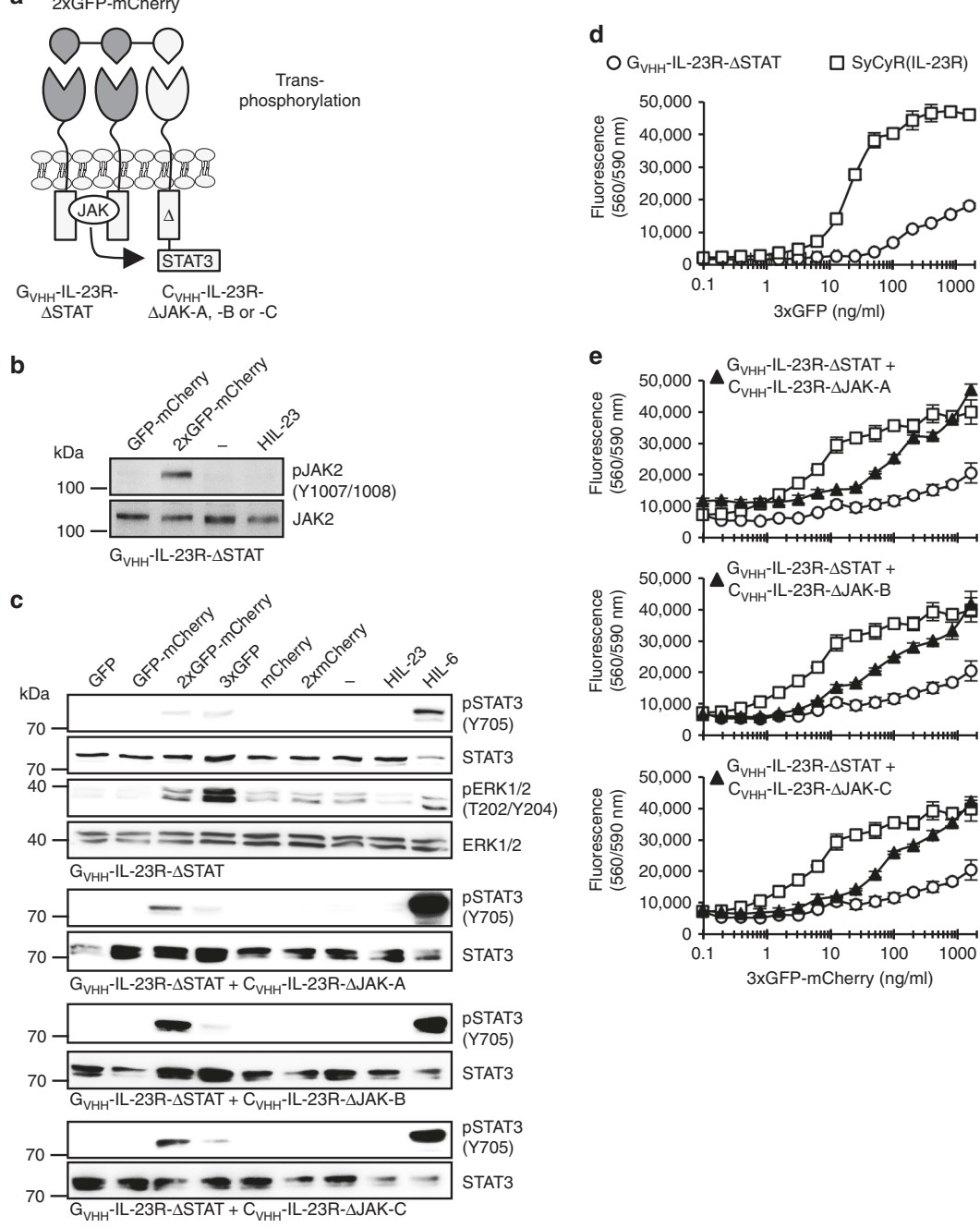

ΔSTAT (Δ503) was amplified by PCR from pcDNA3.1-IL-23R-ΔSTAT (Δ503)[18] and inserted into pcDNA3.1-G$_{VHH}$-IL-23R, where the sequence for IL-23R was removed. Accordingly, cDNAs coding for ΔJAK-A (IL-23R-Δ403–417), ΔJAK-B (IL-23R-Δ455–479), and ΔJAK-C (IL-23R-Δ403-479) were amplified by PCR from p409-IL-23R-Δ403–417, -IL-23R-Δ455–479, and -IL-23R-Δ403-479[20] containing 16 aa of the extracellular domain, the transmembrane domain and the shortened cytoplasmic part of the receptor and inserted into pcDNA3.1-C$_{VHH}$-IL-23R, where the coding sequence for IL-23R was removed. p409-gp130 (chemically synthesized by GeneArt, Thermo Fisher Scientific, Waltham, MA, USA) was digested by EcoRI, NotI and ligated in pcDNA3.1-G$_{VHH}$-IL-23R, which was digested by the same enzymes to generate pcDNA3.1-G$_{VHH}$-gp130 containing 16 aa of the extracellular domain, the transmembrane domain and the intracellular domain of the human receptor. cDNA coding for gp130-ΔSTAT was amplified by PCR (aa 1–758) and ligated to the same vector mentioned before. The expression cassette for 3xGFP was obtained from pmEGFP-13 (Addgene, Cambridge, MA, USA)[46] and inserted into pcDNA3.1 expression vector containing the IL-11R signal peptide and an N-terminal Flag-tag (DYKDDDDK). To generate pcDNA3.1-2xGFP-mCherry, one GFP from pcDNA3.1-3xGFP was removed and replaced with mCherry from pcDNA3.1-mCherry. To create single GFP, the cDNA coding for GFP was amplified by PCR from pcDNA3.1-2xGFP-mCherry and inserted into the pcDNA3.1 expression vector. To create pcDNA3.1-GFP-mCherry, cDNA coding for mCherry was inserted into pcDNA3.1-GFP. For retroviral transduction of Ba/F3-gp130 cells two retroviral plasmids with different resistance genes, pMOWS-puro[43] coding for puromycin resistance and pMOWS-hygro[4] for hygromycin B resistance, have been used. Expression cassettes coding for G$_{VHH}$-IL-23R, C$_{VHH}$-IL-23R-2A-G$_{VHH}$-IL-12Rβ1(SyCyR(IL-23/2A)), C$_{VHH}$-IL-23R, C$_{VHH}$-IL-23R-ΔJAK-A (Δ403–417), C$_{VHH}$-IL-23R-ΔJAK-B (Δ455–479), C$_{VHH}$-IL-23R-ΔJAK-C (Δ403-479), and G$_{VHH}$-gp130 were inserted into pMOWS-puro, whereas those for G$_{VHH}$-IL-12Rβ1, G$_{VHH}$-IL-23R-ΔSTAT (Δ503) and G$_{VHH}$-gp130-ΔSTAT were inserted into pMOWS-hygro. All generated expression plasmids have been verified by sequencing.

**Transfection and selection of cells**. Transfection of U4C and CHO-K1 cells with indicated plasmids was performed using TurboFect™ (Thermo Fisher Scientific, Waltham, MA, USA). Ba/F3-gp130 cells were retrovirally transduced with the pMOWS expression plasmids coding for indicated synthetic receptor variants[18]. Transduced cells were grown in standard DMEM medium as described above supplemented with 10 ng/ml HIL-6. Selection of transduced Ba/F3 cells was performed with puromycin (1.5 µg/ml) or hygromycin B (1 mg/ml) (Carl Roth, Karlsruhe, Germany) or both for at least 2 weeks. Afterwards, HIL-6 was washed away and the generated Ba/F3-gp130 cell lines were selected for GFP:mCherry-dependent growth and analyzed for receptor cell surface expression. Stable transfected CHO-K1 cells secreting GFP:mCherry proteins were selected with 1.125 mg/ml G-418 sulfate (Genaxxon, Biosciences, Ulm, Germany). High-expressing cell clones were identified by western blotting.

**Expression and purification of G$_{VHH}$-C$_{VHH}$ from E.coli**. The cDNA coding for the bispecific antibody G$_{VHH}$-C$_{VHH}$ was generated and subcloned in pet23a. The resulting bispecific antibody sequence was flanked by an N-terminal PelB leader sequence for periplasmic expression and a 3' hexahistidine sequence for purification. Proteins were expressed in the E.coli strain BL21-Rosetta. Bacteria were incubated in 2 l LB-media containing ampicillin 1:1000 (100 µg/ml) and chloramphenicol 1:1000 (34 µg/ml) at 37 °C, until optical density reached 0.6–0.9. Then 1 mM IPTG was added. Bacteria were harvested by centrifugation (5000 × g, 30 min, 4 °C) 4 h after IPTG induction. A cOmplete protease inhibitor tablet (Roche, Mannheim, Germany) was added and supernatant was filtered through a 0.45 µm

bottle top filter. Proteins were purified via IMAC chromatography and eluted with 500 mM imidazole.

**Cell viability assay**. To remove the cytokines, Ba/F3-gp130 cell lines were washed three times with sterile PBS. In all, $5 \times 10^3$ cells were suspended in DMEM supplemented with 10% FCS, 60 mg/l penicillin and 100 mg/l streptomycin and cultured for 3 days in a final volume of 100 µl with or without cytokines/fluorescent proteins as indicated. The CellTiter-Blue Cell Viability Assay (Promega, Karlsruhe, Germany) was used to estimate the number of viable cells by recording the fluorescence (excitation 560 nm, emission 590 nm) using the Infinite M200 PRO plate reader (Tecan, Crailsheim, Germany) immediately after adding 20 µl of reagent per well (time point 0) and up to 2 h after incubation under standard cell culture conditions. All of the values were measured in triplicate per experiment. Fluorescence values were normalized by subtraction of time point 0 values.

**Stimulation assays**. For analysis of STAT3, ERK1/2 and AKT, JAK1, JAK2, and TYK2 activation Ba/F3-gp130 cell lines expressing various SyCyR variants were washed three times with sterile PBS and incubated in serum-free DMEM for at least 2 h. Cells were stimulated with conditioned supernatants of GFP:mCherry fusion proteins as indicated, harvested, frozen in liquid nitrogen, and lysed. Protein concentration of cell lysates was determined by BCA Protein Assay (Pierce, Thermo Fisher Scientific,Waltham, MA, USA). Analysis of STAT3, ERK1/2, AKT, JAK1, JAK2, and TYK2 activation, SOCS3 and β-actin expression was done by immunoblotting using 25–75 µg proteins from total cell lysates and detection with phospho-STAT3 (Tyr705) (D3A7) (1:1000, cat. #9145), phospho-ERK1/2 (Thr-202/Tyr-204) (D13.14.4E) (1:1000, cat. #4370), phospho-AKT (Ser473) (D9E) (1:000, cat. #4060), phospho-JAK1 (Tyr1022/1023) (1:000, cat. #3331), phospho-JAK2 (Tyr1007/1008) (1:000, cat. #3771), phospho-TYK2 (Tyr1054/1055) (1:000, cat. #9321), SOCS3 (C204) (1:500, cat. JP18391) and β-actin (C4) (1:500, cat. sc-47778) mAbs.

**Western blotting**. Defined amounts of proteins from cell lysates were loaded per lane, separated by sodium dodecyl sulfate polyacrylamide gel electrophoresis under reducing conditions and transferred to polyvinylidene fluoride (PVDF) membranes (Carl Roth, Karlsruhe, Germany). The membranes were blocked in 5% fat-free dried skimmed milk (Carl Roth, Karlsruhe, Germany) in TBS-T (10 mM Tris-HCl (Carl Roth, Karlsruhe, Germany) pH 7.6, 150 mM NaCl (AppliChem, Darmstadt, Germany), 1% Tween 20 (Sigma Aldrich, Munich, Germany)) and probed with the indicated primary antibodies in 5% fat-free dried skimmed milk in TBS-T (STAT3 (124H6) (1:1000, cat. #9139), γ-tubulin (1:5000, cat. T5326), GFP (4B10) (1:000, cat. #2955) mAbs) or 5% BSA (Carl Roth, Karlsruhe, Germany) in TBS-T (phospho-STAT3 (Tyr705) (D3A7) (1:1000, cat. #9145), phospho-ERK1/2 (Thr-202/Tyr-204) (D13.14.4E) (1:1000, cat. #4370), ERK1/2 (1:1000, cat. #9102), phospho-AKT (Ser473) (D9E) (1:000, cat. #4060), AKT (1:1000, cat. #9272S), myc (71D10), (1:1000, cat. #2278) and Flag (DYKDDDDK) (1:1000, cat. F7425), SOCS3 (C204) (1:500, cat. JP18391), phospho-JAK1 (Tyr1022/1023) (1:000, cat. #3331), JAK1 (6G4) (1:1000, cat. #3344S), phospho-JAK2 (Tyr1007/1008) (1:1000, cat. #3771), JAK2 (D2E12) (1:000, cat. #3230), phospho-TYK2 (Tyr1054/1055) (1:000, cat. #9321), TYK2 (1:000, cat. #9312) and β-actin (C4) (1:500, cat. sc-47778) mAbs) at 4 °C overnight. After washing, the membranes were incubated with secondary peroxidase-conjugated antibodies (Thermo Fisher Scientific, Waltham, MA, USA, cat. 31462, cat. 31451) 1:2500 diluted in 5% fat-free dried skimmed milk in TBS-T for 1 h at room temperature. PageRuler Prestained Protein Ladder (Thermo Fisher Scientific, Waltham, MA, USA, cat. #26616) was used as MW Marker. The Immobilon™ Western Chemiluminescent HRP Substrate (Merck Chemicals GmbH, Darmstadt, Germany) and the ChemoCam Imager (INTAS Science

**Fig. 7** Engineered heterotrimeric SyCyRs of IL-23R are capable of STAT3 trans-phosphorylation in transduced Ba/F3-gp130 cells. **a** Schematic illustration of IL-23R-SyCyRs simulating STAT3 trans-phosphorylation. The 2xGFP-mCherry fusion protein served as synthetic cytokine ligand. G$_{VHH}$-IL-23R-ΔSTAT consists of the GFP-nanobody fused to 16 aa of the extracellular part, the transmembrane and intracellular domains of the IL-23R lacking STAT-binding motifs. The C$_{VHH}$-IL-23R-ΔJAK variants consist of the mCherry-nanobody fused to 16 aa of the extracellular part, the transmembrane and intracellular domains of the IL-23R lacking the JAK activation site (ΔJAK-A,-B,-C). **b** Analysis of JAK activation in Ba/F3-IL-23R-ΔSTAT cells. Cells were washed three times, starved, and stimulated with 100 ng/ml of the indicated synthetic ligands for 20 min. Cellular lysates were prepared, and equal amounts of total protein (50 µg/lane) were loaded on SDS gels, followed by immunoblotting using specific antibodies for phospho-JAK2 and JAK2. Western blot data show one representative experiment out of three. **c** Analysis of signal transduction in Ba/F3-G$_{VHH}$-IL-23R-ΔSTAT cells and Ba/F3-G$_{VHH}$-IL-23R-ΔSTAT cells stably transduced with one of the C$_{VHH}$-variants depicted in **a**. Ba/F3-G$_{VHH}$-IL-23R-ΔSTAT cells and variants thereof were washed three times, starved, stimulated with 4% CHO-K1 conditioned supernatant containing mCherry, 2xmCherry or conditioned supernatant containing 100 ng/ml of the other indicated synthetic ligands for 60 min. Stimulation with HIL-23 (10 ng/ml) for 60 min and HIL-6 (10 ng/ml) for 15 min was used as control. Cellular lysates were prepared, and equal amounts of total protein (for STAT3 analysis 25 µg/lane for HIL-6 and 50 µg/lane for other ligands, for ERK1/2 analysis 12.5 µg/lane for HIL-6 and 75 µg/lane for other ligands) were loaded on SDS gels, followed by immunoblotting using specific antibodies for phospho-STAT3/ERK1/2 and STAT3/ERK1/2. Western blot data show one representative experiment out of two. **d**, **e** Cellular proliferation of Ba/F3-SyCyR(IL-23R), Ba/F3-G$_{VHH}$-IL-23RΔSTAT cells and variants thereof with 3xGFP or 2xGFP-mCherry fusion proteins. Equal numbers of cells were cultured for 3 days in the presence of the indicated ligands (0.1–1600 ng/ml). Proliferation was measured using the colorimetric CellTiter-Blue cell viability assay. One representative experiment out of four is shown. Results are mean ± s.d. of three replicates

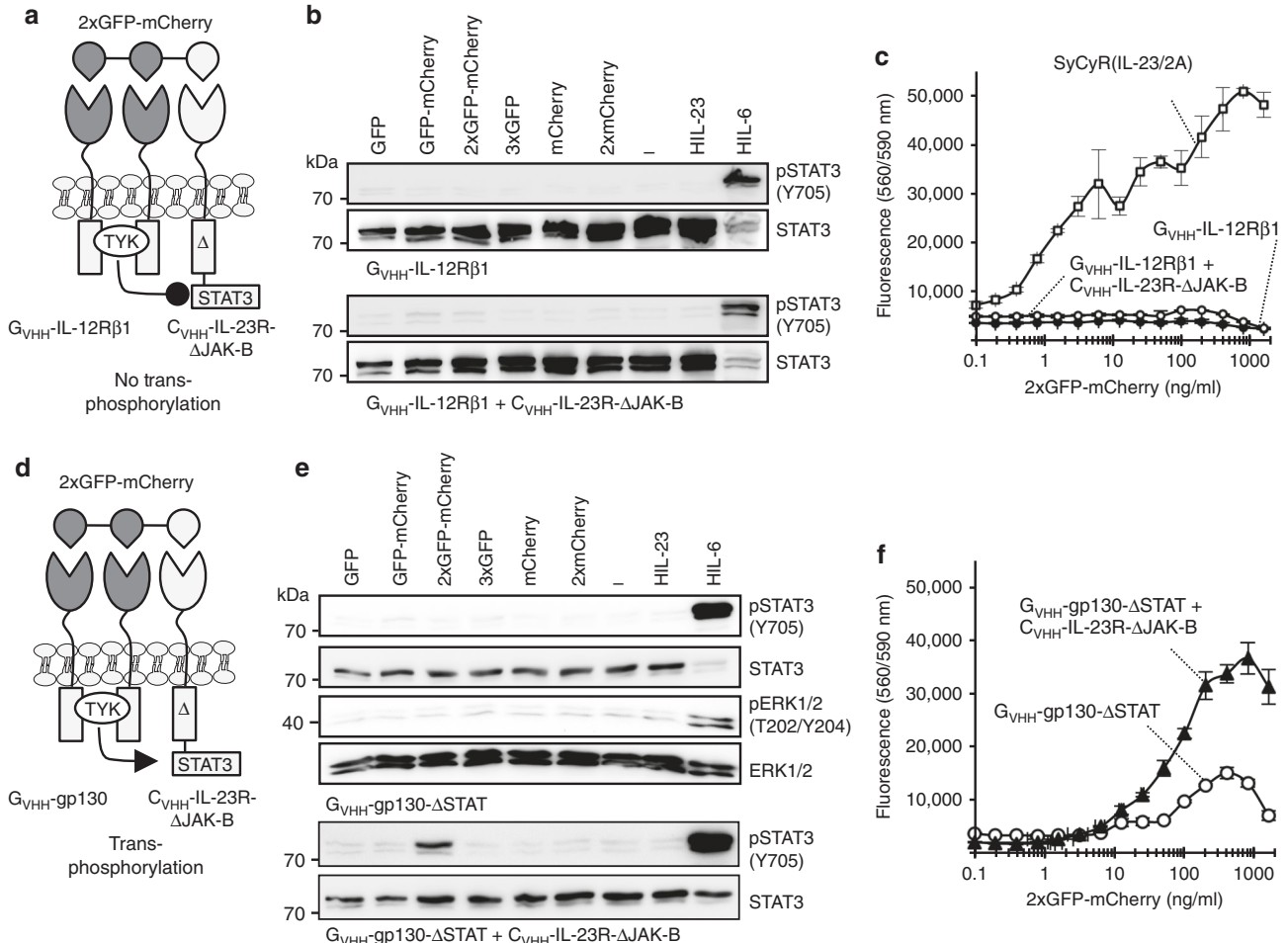

**Fig. 8** Engineered heterotrimeric SyCyRs based on $G_{VHH}$-IL-12Rβ1 and $G_{VHH}$-gp130 homodimerization fail in STAT3 trans-phosphorylation. **a** Schematic illustration of IL-12Rβ1-SyCyRs simulating STAT3 trans-phosphorylation using 2xGFP-mCherry. $G_{VHH}$-IL-12Rβ1: GFP-nanobody fused to 15 aa of ECD, TMD, and ICD of IL-12Rβ1, lacking any STAT-binding motifs. $C_{VHH}$-IL-23R-ΔJAK-B variant: mCherry-nanobody fused to 16 aa of ECD, the TMD and ICD of the IL-23R lacking JAK activation site (ΔJAK-B). **b** STAT3 activation in indicated Ba/F3-$G_{VHH}$-IL-12Rβ1 cells lines were stimulated with 16% CHO-K1 conditioned supernatant containing mCherry, 2xmCherry or conditioned supernatant containing 400 ng/ml of the other indicated ligands for 60 min or with HIL-23 (10 ng/ml) for 60 min and HIL-6 (10 ng/ml) for 15 min. Total proteins (25 μg/lane for HIL-6 and 50 μg/lane for other ligands) were analyzed for phospho-STAT3 and STAT3. Western blot data show one representative experiment out of three. **c** Proliferation of indicated Ba/F3-SyCyR(IL-23/2A) and Ba/F3-$G_{VHH}$-IL-12Rβ1 cell lines were cultured in the presence of 2xGFP-mCherry (0.1–1600 ng/ml). One representative experiment out of two is shown. Results are mean ± s.d. of three replicates. **d** Schematic illustration of gp130-SyCyRs simulating STAT3 trans-phosphorylation with 2xGFP-mCherry. $G_{VHH}$-gp130-ΔSTAT: GFP-nanobody fused to 13 aa of the ECD, the TMD and ICD of gp130 lacking the STAT-binding motifs. $C_{VHH}$-IL-23R-ΔJAK-B: mCherry-nanobody fused to 16 aa of the ECD, the TMD and ICD of the IL-23R lacking JAK activation site. **e** STAT3 and ERK1/2 activation in indicated Ba/F3-$G_{VHH}$-gp130-ΔSTAT cell lines were stimulated with 8% CHO-K1 conditioned supernatant containing mCherry, 2xmCherry or conditioned supernatant containing 200 ng/ml of the other indicated ligands for 60 min or HIL-23 (10 ng/ml) for 60 min or HIL-6 (10 ng/ml) for 15 min. Equal amounts of total protein (for STAT3 analysis 25 μg/lane for HIL-6 and 50 μg/lane for other ligands. For ERK1/2 analysis 12.5 μg/lane for HIL-6 and 75 μg/lane for other ligands) were analyzed for phospho-STAT3/ERK1/2 and STAT3/ERK1/2. Western blot data show one representative experiment out of two. **f** Proliferation of indicated Ba/F3-$G_{VHH}$-gp130-ΔSTAT cells lines were cultured in the presence of 2xGFP-mCherry (0.1–1600 ng/ml). One representative experiment out of four is shown. Results are mean ± s.d. of three replicates

Imaging Instruments GmbH, Göttingen, Germany) were used for signal detection. For re-probing with another primary antibody, the membranes were stripped in 62.5 mM Tris-HCl (Carl Roth, Karlsruhe, Germany) pH 6.8, 2% SDS (Carl Roth, Karlsruhe, Germany) and 0.1% β-mercaptoethanol (Sigma Aldrich, Munich, Germany) for 30 min at 60 °C and blocked again.

The band intensities of the western blots (Fig. 4f) were quantified using ImageJ software. Uncropped images of western blots are presented in Supplementary Fig. 13.

**Cell surface detection of cytokine receptors**. To detect cell surface expression of the synthetic cytokine receptors, stably transduced Ba/F3-gp130 cells were washed with FACS buffer (PBS containing 1% BSA) and incubated at $5 \times 10^5$ cells/100 μl FACS buffer supplemented with a 1:100 dilution of anti-myc (71D10), (cat. #2278) or 1.2 μg anti-Flag (DYKDDDDK) (cat. F7425) mAbs for 1 h on ice. After a single

wash with FACS buffer, cells were incubated in 100 μl FACS buffer containing a 1:100 dilution of Alexa Fluor 488 conjugated Fab goat anti-rabbit IgG (cat. A11070) for 1 h on ice. Finally, cells were washed once with FACS buffer, suspended in 500 μl FACS buffer and analyzed by flow cytometry (BD FACSCanto II flow cytometer, BD Biosciences, San Jose, CA, USA). Data was evaluated using the FCS Express 4 Flow software (De Novo Software, Los Angeles, CA, USA).

**Microarray analysis**. Ba/F3-gp130 cells were grown in DMEM high-glucose culture medium supplemented with 10% fetal calf serum (GIBCO®, Life Technologies, Darmstadt, Germany), 60 mg/l penicillin and 100 mg/l streptomycin (Genaxxon Bioscience GmbH, Ulm, Germany) at 37 °C with 5% $CO_2$ in a water saturated atmosphere. In all, 10 ng/ml of conditioned cell culture medium from a stable CHO-K1 clone secreting Hyper-IL-6 (stock solution approx. 5 μg/ml as determined by ELISA) were used to supplement the growth medium. Ba/F3-gp130 cells were

stable transduced with different receptor complexes (IL-12Rβ1-IL-23R, C$_{VHH}$-IL-23R-2A-G$_{VHH}$-IL-12Rβ1(SyCyR(IL-23/2A)) or G$_{VHH}$-IL-23R (SyCyR(IL-23R))), independently selected and cultivated for several weeks. Subsequently, Ba/F3-gp130 cell lines were washed four times with sterile PBS and incubated in serum-free DMEM for 3 h. Equal numbers of cells ($2 \times 10^6$) were stimulated with 100 ng/ml HIL-23, GFP-mCherry or 3xGFP for 1 h at 37 °C, independently. Stimulation with cell culture supernatant from untransfected CHO-K1 cells was used as control. Total RNA extraction of four independent biological replicates was made with RNeasy Mini Kit (Qiagen, Hilden, Germany) according to the manufacturer's instructions. RNA quality was evaluated using an Agilent 2100 Bioanalyzer and only high-quality RNA (RIN > 8) was used for microarray analysis. For this, total RNA (150 ng) was processed using the Ambion WT Expression Kit and the WT Terminal Labeling Kit (Thermo Fisher Scientific, Waltham, MA, USA) and hybridized on Affymetrix Mouse Gene ST 1.0 arrays containing about 28,000 probe sets. Staining and scanning were done according to the Affymetrix expression protocol. Expression console (Affymetrix, Freiburg, Germany) was used for quality control and to obtain annotated normalized RMA gene-level data (standard settings including sketch-quantile normalization). Statistical analyses were performed by utilizing the statistical programming environment R (R Development Core Team[47]) implemented in CARMAweb[48] (1.5-fold, $p$-value ≤ 0.01). Data were analyzed pairwise, Ba/F3-SyCyR(IL-23/2A) cells stimulated with vs. without 100 ng/ml GFP-mCherry and Ba/F3-IL-12Rβ1-IL-23R cells stimulated with vs. without 100 ng/ml HIL-23 and Ba/F3-SyCyR(IL-23R) cells stimulated with vs. without 100 ng/ml 3xGFP .

GO term and pathway enrichment analyses ($p < 0.01$ of enrichment) of differential abundant transcripts (1.5-fold, $p$-value ≤ 0.01) were done with Ingenuity software (Qiagen, Hilden, Germany). Gene expression raw data are available at GEO (accession number GSE101569).

**Animals**. C57BL/6 mice (Janvier Labs) were obtained from the animal facility of the University of Düsseldorf. Mice were fed with a standard laboratory diet and given autoclaved tap water ad libitum. They were kept in an air-conditioned room with controlled temperature (20–24 °C), humidity (45–65%), and day/night cycle (12 h light, 12 h dark). Mice were acclimatized for 1 week before entering the study. All procedures were performed in accordance with the national guidelines for animal care and were approved by the local Research Board for animal experimentation (LANUV, State Agency for Nature, Environment and Consumer Protection, approval number (Az. 84-02.04.2016.A025)).

**Hydrodynamic-based in vivo gene delivery**. Eight-week-old male C57BL/6 mice were transfected via tail vein injection of the plasmids pcDNA3.1-G$_{VHH}$-gp130 (2.3 µg/mouse) and/or pcDNA3.1-3xGFP (1.28 µg/mouse) prepared in PBS as described previously[49]. Animals were allocated randomly to the various groups. The studies were sufficiently powered based on power calculation at α of 0.05 and β of 0.2.

**Gene expression analysis of Saa1**. RNA was isolated with TRIzol according to the manufacturer's instructions (Thermo Fisher Scientific, Waltham, MA, USA). Gene expression of Saa1 (QT00196623, Qiagen, Hilden, Germany) and Gapdh (forward: TGCACCACCAACTGCTTAG, reverse: GGATGCAGGGATGATGTTC) were performed using a One-step SYBR Kit. (Bio-Rad Laboratories, Hercules, CA, USA).

**Preparation of liver lysates**. Tissue protein extracts from liver were prepared on ice using the lysis buffer (50 mM Tris-HCl (Carl Roth, Karlsruhe, Germany), 150 mM NaCl (AppliChem, Darmstadt, Germany), 2 mM EDTA (Sigma Aldrich, Munich, Germany), 2 mM NaF (Sigma Aldrich, Munich, Germany), 1 mM Na$_3$VO$_4$ (Sigma Aldrich, Munich, Germany), 1% Nonidet P40 BioChemica (AppliChem, Darmstadt, Germany) 1% Triton X-100 (Sigma Aldrich, Munich, Germany) and cOmplete EDTA-free Protease inhibitor cocktail tablet (Roche Diagnostics, Mannheim, Germany) and analyzed by western blotting. Equal amounts of protein (50 µg/lane) were loaded. Western blotting was scored by an observer blinded to the groups.

**Statistical analysis**. Data are presented as mean ± SEM. The Brown–Forsythe test was used to examine homogeneity or equality of variance. For multiple comparisons, one-way ANOVA, followed by Bonferroni post hoc tests, was used (GraphPad Prism 6.0, GraphPad Software Inc., San Diego, CA, USA). Statistical significance was set at the level of $p < 0.05$.

**Data availability**. The authors declare that the data supporting the findings of this study are available within the paper and from the authors on request. Gene expression raw data that support the findings of this study have been deposited in Gene Expression Omnibus (GEO) with the accession number GSE101569.

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

## Acknowledgements

We thank Petra Oprée for assistance. This work was funded by grants from the Deutsche Forschungsgemeinschaft (RTG1949, SFB974 and SFB 1116).

## Author contributions

E.E., A.S. conducted most of the experiments and analyzed the data. M.F., R.C., A.L., P.B., J.M.M., and D.M.F. supported cloning, recombinant protein expression and cell culture experiments. H.X., P.A.L. performed the in vivo mice study. C.B., B.K., H.A.H. performed the microarray study. All authors helped writing the paper. J.S. designed the study, analyzed the data, and wrote the paper.

## Additional information

**Competing interests:** The authors declare no competing interests.

