## [Peer Review File · Nature Communications]

Reviewers' comments:

Reviewer #1 (Remarks to the Author):

This is a nice system to generate artificial chimeric cytokine receptors using Nanobodies to GFP and mCherry. I have no faults with the paper just that there are many many examples of such chimeric receptor systems, including those listed by the authors, that have been used to analyze cytokine receptor systems so I don't think the novelty rises to Nature Communications. While I understand that this system is "switchable" there are other systems out there that would also make that claim using different approaches. Again, it's nice work and will be useful, but I question the novelty.

Reviewer #2 (Remarks to the Author):

This is a well-constructed and clever approach to dissecting out the basics of cytokine signaling. It was easy to read and flowed well and, importantly, holds a lot of potential for future basic cytokine biology studies. It is also entirely possible to see this combined with engineered CAR T cells to allow really specific modulation of these cells in the setting of cancer and so there is a very real translational potential.

Minor comments

Hatching in Fig 1B legend is hard to distinguish mCherry and GFP-mCherry.

For Fig 2A – it does look like STAT3 phosphorylation is declining with time. Although this is not targeted by SOCS3 there are other ways to limit this signaling. Perhaps the authors could change permanent to sustained?

Figure 3 – the genes induced by the synthetic and natural ligand have an 85% overlap (the main point) but also implies there is a 15% difference. From the description of the experiment it is unclear how many times this was performed and how many replicates were used in each experiment. It would not be surprising to see 5-10% variation between replicate samples so it is unclear if the 15% variation they see is real or simply noise in the system. There are a couple of ways to deal with this – increase the threshold for change and/or take a hard look at the genes that are different and see if those that are different are related etc. This is a minor point.

Figure 5 – sort of surprising that 23R homo-dimerization would have this activity – but interesting to see. This is not essential – but did the authors ever try and engineer this system to just express the full length 23R and determine if IL-23, p40 or p19 alone was sufficient to send a similar signal?

Reviewers' comments:**Reviewer #1 (Remarks to the Author):****Reviewer comment:**

This is a nice system to generate artificial chimeric cytokine receptors using Nanobodies to GFP and mCherry. I have not faults with the paper just that there are many many examples of such chimeric receptor systems, including those listed by the authors, that have been used to analyze cytokine receptor systems so I dont think the novelty rises to Nature Communications. While I understand that this system is "switchable" there are other systems out there that would also make that claim using different approaches. Again, its nice work and will be useful, but I question the novelty.

Authors answer:

Our system can be adapted to a wide range of cytokine receptors, here exemplified for gp130 and IL-23R signalling. It can be easily imagined that this system will be of high interest for the recently developed CAR-technology to selectively activate transgenic T-cells with synthetic receptors to prolong life span or proliferation rate. Moreover transgenic mice carrying these receptors under tissue specific promoters or the CRE-lox system will be of high relevance to analyse the activity of the respective cytokine on a certain cell type without affecting other cells in the body. This is simply not possible with any other system using natural cytokines or other synthetic cytokine receptors as we have discussed and thus signifies the novelty of our approach.

The beauty of this synthetic receptor system is its simplicity. All proteins and cDNAs are easily produced in the lab and many scientists have long standing experience with fluorescent proteins and antibodies lowering their inhibition threshold. All scientists that I have presented this system were enthusiastic and immediately started to plan and conduct experiments, that were simply not possible before. So I expect that this system will be used quickly worldwide in research of cytokine biology. Thus, while chimeric receptor systems have been used before, our system presents an important new development for synthetic biology.

Reviewer #2 (Remarks to the Author):

This is a well-constructed and clever approach to dissecting out the basics of cytokine signaling. It was easy to read and flowed well and, importantly, holds a lot of potential for future basic cytokine biology studies. It is also entirely possible to see this combined with engineered CAR T cells to allow really specific modulation of these cells in the setting of cancer and so there is a very real translational potential.

Minor comments

Reviewer comment:

Hatching in Fig 1B legend is hard to distinguish mCherry and GFP-mCherry.

Authors answer:

The reviewer is right. The Figure 1B (and 5B) is now presented with a color code.

Reviewer comment:

For Fig 2A – it does look like STAT3 phosphorylation is declining with time. Although this is not targeted by SOCS3 there are other ways to limit this signaling. Perhaps the authors could change permanent to sustained?

Authors answer:

We have changed the term „permanent“ into „sustained“.

Reviewer comment:

Figure 3 – the genes induced by the synthetic and natural ligand have an 85% overlap (the main point) but also implies there is a 15% difference. From the description of the experiment it is unclear how many times this was performed and how many replicates were used in each experiment. It would not be surprising to see 5-10% variation between replicate samples so it is unclear if the 15% variation they see is real or simply noise in the system. There are a couple of ways to deal with this – increase the threshold for change and/or take a hard look at the genes that are different and see if those that are different are related etc. This is a minor point.

Authors answer:

Overall, the holistic gene expression experiments were performed to confirm, that the synthetic and natural IL-23 signaling systems adequately target downstream signaling components.

Importantly, we performed transcriptome analysis using samples of four independent biological replicates. Experiments were performed using two different cell systems and various ligands for cell stimulation. This further means, that Ba/F3-gp130 cells were grown in DMEM^{+/+} (conditioned 60 mg/l penicillin and 100mg/l streptomycin and 10% FCS) with Hyper-IL-6 (0.2% cell culture supernatant from CHO-K1 cells), stable transduced with different receptor complexes (IL-12R β 1-IL-23R or C_{VHH}-IL-23R-2A-G_{VHH}-IL-12R β 1(SyCyR(IL-23/2A))), independently selected and cultivated for several weeks. Subsequently, Ba/F3-gp130 cell lines were washed four times with sterile PBS and incubated in serum-free DMEM for 3 h. Equal numbers of cells (2×10^6) were stimulated with 100 ng/ml HIL-23 or GFP-mCherry for 1 h at 37°C, independently. Stimulation with cell culture supernatant from untransfected CHO-K1 cells was used as control. Total RNA extraction was made with RNeasy Mini Kit (Qiagen, Hilden, Germany) according to the manufacturer's instructions.

To address the important point raised by the reviewer, we added the information to the methods section:

“In brief, Ba/F3-gp130 cells were grown in DMEM high glucose culture medium supplemented with 10% fetal calf serum (GIBCO®, Life Technologies), 60 mg/l penicillin and 100 mg/l streptomycin (Genaxxon Bioscience GmbH, Ulm, Germany) at 37°C with 5% CO₂ in a water saturated atmosphere. 10 ng/ml of conditioned cell culture medium from a stable CHO-K1 clone secreting Hyper-IL-6 (stock solution approx. 5 μ g/ml as determined by ELISA) were used to supplement the growth medium. Ba/F3-gp130 cells were stable transduced with different receptor complexes (IL-12R β 1-IL-23R, C_{VHH}-IL-23R-2A-G_{VHH}-IL-12R β 1(SyCyR(IL-23/2A)) or G_{VHH}-IL-23R (SyCyR(IL-23R)), independently selected and cultivated for several weeks. Subsequently, Ba/F3-gp130 cell

lines were washed four times with sterile PBS and incubated in serum-free DMEM for 3 h. Equal numbers of cells (2×10^6) were stimulated with 100 ng/ml HIL-23, GFP-mCherry or 3xGFP for 1 h at 37°C, independently. Stimulation with cell culture supernatant from untransfected CHO-K1 cells was used as control. Total RNA extraction of four independent biological replicates was made with RNeasy Mini Kit (Qiagen, Hilden, Germany) according to the manufacturer's instructions".

In regard to our hypothesis, the most important finding from the microarray analysis was that of the overall approximately 28,000 probes sets on the Array used, only such a small subset of transcripts was differently abundant at all.

Of these differential abundant transcripts, almost all natural ligand targets were also identified to be regulated by the synthetic ligand, using our significance levels (1.5-fold, p-value ≤ 0.01).

For our surprise, we could not detect 15% of transcripts for the synthetic system compared to the natural system. Nevertheless, by relaxing the statistical significance levels (1.3-fold, p-value ≤ 0.05) for most of these transcripts ($\sim 10\%$) the trend can be confirmed, as targets were regulated in the same direction in the synthetic system as in the natural system. For the remaining $\sim 5\%$ of the transcripts this trend could not be confirmed.

Reviewer comment:

Figure 5 – sort of surprising that 23R homo-dimerization would have this activity – but interesting to see. This is not essential – but did the authors ever try and engineer this system to just express the full length 23R and determine if IL-23, p40 or p19 alone was sufficient to send a similar signal?

Authors answer:

Our unpublished, preliminary data show that IL-23 proteins form homodimers (co-immunoprecipitation) and these homodimers are able to induce signaling at higher concentrations (>10 ng/ml) most likely via a IL-23R homodimer (on Ba/F3 cells only expressing IL-23R). We are, however, still working on this project and our current view is that IL-23 is fully biologically active as homodimer. We have included this data as additional information "for reviewers only".

REVIEWERS' COMMENTS:

Reviewer #2 (Remarks to the Author):

Well reasoned responses and questions directly addressed.